# Framework and Methodology for Establishing Port-City Policies Based on Real-Time Composite Indicators and IoT: A Practical Use-Case

**DOI:** 10.3390/s20154131

**Published:** 2020-07-24

**Authors:** Ignacio Lacalle, Andreu Belsa, Rafael Vaño, Carlos E. Palau

**Affiliations:** Departamento de Comunicaciones, Universitat Politècnica de València, 46022 Valencia, Spain; anbelpel@upv.es (A.B.); ravagar2@upv.es (R.V.); cpalau@dcom.upv.es (C.E.P.)

**Keywords:** Smart Port-Cities, composite indicator, real-time, Internet of Things, traffic congestion, framework and methodology

## Abstract

During the past few decades, the combination of flourishing maritime commerce and urban population increases has made port-cities face several challenges. Smart Port-Cities of the future will take advantage of the newest IoT technologies to tackle those challenges in a joint fashion from both the city and port side. A specific matter of interest in this work is how to obtain reliable, measurable indicators to establish port-city policies for mutual benefit. This paper proposes an IoT-based software framework, accompanied with a methodology for defining, calculating, and predicting composite indicators that represent real-world phenomena in the context of a Smart Port-City. This paper envisions, develops, and deploys the framework on a real use-case as a practice experiment. The experiment consists of deploying a composite index for monitoring traffic congestion at the port-city interface in Thessaloniki (Greece). Results were aligned with the expectations, validated through nine scenarios, concluding with delivery of a useful tool for interested actors at Smart Port-Cities to work over and build policies upon.

## 1. Introduction

Traditionally, port and cities have been working as independent siloes, striving separately to reach innovation goals towards better sustainability and economic growth. During the past few decades, the increasing pace of urban population growth [1] and expansion of maritime commerce [2] have brought forth environmental and societal challenges which affect both sides. In light of new wave technologies that are already having an impact on port-cities, the moment to tackle joint challenges making use of the most advanced innovations towards the future Smart Port-Cities is now [3]. Looking only at the big numbers, there are 33 megacities in the world (with more than 10 million inhabitants) [4]; 25 of them are port-cities, accumulating more than 25% of global container trade volume [5]. Naturally then, useful solutions addressing Smart Port-City challenges will be a key game changer from both business and societal perspective in the forthcoming years. This is the context in which this work is framed.

A Smart Port-City is, by definition, the combination of a Smart City with a Smart Port. The notion dates back to the SmartPORT initiative proposed by Hamburg in 2012 [6]. However, the concept goes far beyond that. A Smart Port-City entails the integration of information from the two domains in a transparent manner and the realization that port-city actions must be envisaged as a holistic entity taking advantage of the era of massive data [7]. One of the main challenges of Smart Port-Cities is the establishment of effective policies between the Port Authorities (and relevant agents) and the Municipality or Regional Governments [3]. Agreeing upon security, mobility, energy and environmental matters appears crucial for the future of port cities. Those policies should lead to a more efficient and productive mutually beneficial relationship between the involved entities. According to the Association Internationale Villes et Ports (AIVP) Agenda 2030 for Sustainable Port-Cities [8], “*port cities are the best scenarios to test innovative solutions tackling different issues*”. AIVP went farther in the list, and clearly defined 10 sustainable goals: climate change, energy transition, sustainable mobility, renewed governance, port culture and identity, port-city interface, health and life quality and protecting biodiversity.

Among the former, a huge concern for citizens, municipalities, and all kinds of organisations (Non-Governmental Organizations—NGOs—included) are those related with vehicle traffic [9] and its effects, which are directly related to climate change. Nowadays, the vast majority of urban vehicles are still powered by non-degradable, highly-polluting fossil fuels [10], with a series of hazardous consequences to the ecosystem and its inhabitants [11]. There is a growing trend to establish norms and policies seeking to reduce the pollution levels at cities; either supported by legislation [12] or just initiatives for improving both traffic congestion and the well-being levels of citizens [13]. Linking with the goal of this work, this effect is even more noticeable in port-cities, where the fact of hosting a port in the immediate surroundings of living neighborhoods entails its own hindrances (e.g., [14]). While business efficiency may drive operational and technical innovations, sometimes supported by Information and Communications Technology—ICT [15], sustainability is also a true stakeholder value for port authorities. Achieving sustainability goals for a port, such as by reducing its emissions or being more transparent on port-city relations can provide a long-term payback. Despite being difficult to measure, this payback can yield immaterial gains such as increased residential citizens’ acceptance of port operations, which would be an ideal pillar to build policies upon. There are documented examples of ports striving through numerous initiatives and investments to raise awareness and improve citizen acceptability [16]. However, today, there are no consolidated indicators to demonstrate to the citizens the strong commitment of port authorities, to reduce the environmental impacts and other potential burdens of their activities. The most relevant reference is the Global Reporting Initiative—GRI—guidelines for sustainability, which does not even have seaport-sector-specific guidelines [17].

Taking advantage of Internet of Things—IoT—techniques and real-time features opens a wide range of options for defining port-cities policies such as enabling decision-making after analysing traffic congestion at specific spots in cities. Allowing the different actors in the port-city relation to measure, analyse and act based on solid, formal, reliable indicators will enable, for instance, setting specific thresholds on the traffic volumes, establishing alarms and alerts over certain ranges, accounting for responsibilities during specific events, proposing alternative mobility solutions or simply checking the evolution throughout time for tackling further actions.

The port-city interface is normally modelled as a single convergence point having the following features: demographics, urban development, public tender procurement, traffic, noise, pollution, employment and sustainability. To drive towards sustainable benefits over that interface, it is worthwhile to follow the principles indicated in the “Sustainable Cities” report by the European Commission [18], which indicates that clear quantified indicators must be used for that purpose. Not only are indicators powerful for measuring progress and performance in specific areas; according to the Policy Influence for Indicators—POINT—study [19], those indicators have a huge potential to drive the creation of policies. To use them as tools for defining policies, indicators must be both quantifiable and measurable. Additionally, the construction of those indicators must be a joint activity of both municipality/regional authorities and the port itself. Co-design of the methodology, significance, relevance, and boundaries of each indicator is a challenging task that must be faced in a joint fashion.

The indicators will be, sometimes, rather simple (e.g., the number of jobs generated by the port). In contrast, more often those indices will be complex due to the influence of different parameters (e.g., the environmental pollution from the port or the traffic increase attributable to port activities). In this work, the authors propose to use composite indicators with that purpose. According to the European Commission, composite indicators are ideal for situations where different parts of an event representation have no common meaningful unit of measurement and there is no obvious way of weighting them to have a single illustrative value [20]. This definition fits perfectly the usual case on Smart Port-Cities interface points. The rationale behind that usage will be to come up with composite indicators to be used to advocate desirable policies, ensuring that the process of construction is transparent, and it has clear backing statistical conceptual principles.

The paper’s objectives can then be summarized as follows:(1)To explore how to translate the knowledge of the field of composite indicators to the port-city interface.(2)To define how to build solid indicators with that purpose, focusing on their calculation and prediction to represent real-world phenomena in the context of a Smart Port-City(3)To analyse the various technological options to construct an architecture leveraging IoT techniques covering the indicator implementation.(4)To propose an IoT-based software framework accompanied with a methodology for its co-design and deployment on a real use-case.(5)To effectively conduct a small scale experiment consisting of implementing a composite indicator use-case reflecting the traffic congestion in the interface between the port and the city of Thessaloniki (Greece).

Underlying those objectives, there appears a hypothesis aim to be validated: “IoT can be used for deploying solid, robust services for assisting the decision-making on setting port-city policies”.

The framework developed has been applied and deployed in a real Smart Port-City (Thessaloniki), thus providing a legitimate testbed. Additionally, the work in this paper has usability potential as an IoT solution for mission-critical infrastructures of the Smart City, as the port and its interface with the city can be considered as such [21]. Besides, the work has advanced in the definition of semantic modeling of city knowledge, specifically in the traffic congestion realm. Finally, one of the last objectives of this work is to apply time-series forecasting over traffic and city data, clearly seeking to extract knowledge using machine learning (ML). In particular, it focuses on both understanding phenomena from historic data and catering to short-term prediction decisions using real-time data.

The reminder of the article is organized as follows: Section 2 presents the state-of-the-art and relevant previous work. Section 3 explains thoroughly the proposed framework and its methodology to be applied in a use-case, which is presented as well. Section 4 shows the results of applying the framework to a real traffic congestion scenario. Section 5 briefly discusses the knowledge to be extracted from the results and its utility. Last, Section 6 concludes the article with some suggestions on future work directions.

## 2. Related Work

### 2.1. Open Data in Smart Ports and Smart Cities

Open data is a key element in the relationship between the Smart Cities and Smart Ports. Reference [22] describes open data initiatives as part of the efforts by governments to provide transparency, better empower citizens, foster innovation, and reform public services by investigating the impacts of open data innovation on different Smart City domains like the economy, education, energy, the environment, governance, tourism, transport and mobility. Reference [23] explains that open data is not only global data collected and opened by the government, but also includes data shared among individual citizens and industries with the government and public. Related with this paper experiment’s domain, reference [24] analyses the current and future situation of open data in Smart Ports. It summarises the benefits of using open data for enabling typical applications in the port operations (e.g., truck gate appointments, vessel calls data management, yard operation scheduling or hazardous material tracking within the port premises), helping coordinate port operations to deal with unexpected delays caused by city issues like traffic congestion and a fluid exchange information between the port community and the port vicinity. However, a reluctancy of ports to release certain data to the community (researchers, other stakeholders) was also observed, especially those that concern business model operations or private protected data.

Nowadays, the main world cities provide their own open data platforms. For example, the European Data Portal has analysed the status and features of Open Data portals in eight of the largest capital cities [25] in Europe and in medium-sized European cities [26] like Thessaloniki. The information from these portals is useful for the ports, as it is originated by sensors connected to the surrounding areas providing flow of information between the entire city, the port neighborhood, and the port. The analysis, use and processing of this data is leading to the advent of applications and services improving the daily activities of the port and their impact on the environment.

### 2.2. Interoperability at Middleware Level of Heterogeneous Data in a City

From an architectural point of view, a study of the current status of functionalities, components and platforms to achieve IoT interoperability that can be used in a Smart Port City solution was performed.

Some research projects focused on achieving interoperability between IoT platforms provide a detailed state of the art overview of the architecture of existing IoT platforms. The INTER-IoT [27] study is based on a layered classification approach regardless of the application domains. This project covered the whole IoT architecture to achieve a complete interoperability solution including devices and platforms. The ACTIVAGE project [28] study is closely related with middleware platforms and is based on a layered classification: device level, connectivity level, cloud level and application level; providing a complete picture of the features and functionalities of the main IoT platforms. After reviewing the literature, the main research needs on that regard for completing this paper are data access from several heterogeneous data sources and open data and the middleware interoperability.

Related to access to the data sources, ref. [29] presents a city-wide IoT data management layer providing some Representational State Transfer—REST—and linked Open Data Application Programming Interfaces—APIs—that collect and show data related to elderly people. That work highlights the need of defining a common format to integrate the data, allowing one to define different levels of data abstraction, scalable infrastructure, and semantic meaning to the stored information. Reference [30] explains an ontology for semantic interoperability and linked data integration. The FIWARE project [31] proposes some data models [32] to enable data portability for different applications highlighting their potential for Smart City solutions. Regarding middleware interoperability, some available open source technologies like Apache Kafka [33] provide real-time processing of IoT events, while some works (e.g., [30]) depict “the sensor middleware” as a main element that collects, filters, and combines data streams from virtual sensors or physical devices. In addition to solutions like Kafka, that approach ensures proper semantic annotations of the data collected. FIWARE provides the ORION Context Broker [34], a component to manage the entire lifecycle of context information in a common format.

Focusing on middleware-level implementations for Smart Cities in the literature, ref. [35] performs an exhaustive analysis of the international organizations involved in Smart City standardisation. It presents a complete study about Smart City vendors and their software systems. It starts by exposing the two barriers to Smart City solutions. First, the current Smart City technologies are based on non- interoperable custom systems that are not replicable in other cities. Second, the architectural design efforts have not yet converged to a clear reference, creating uncertainly among stakeholders. It compares 47 different Smart City software systems based in their domain work, communication protocols, training and promotion resources, location of deployments and open software design and architecture. It emphasizes the importance of the methodology to choose a Smart City software architecture system. Reference [36] lists some platforms and solutions from a design and architectural point of view classifying them by their requirement analysis, challenges achieved and reference architecture and comparing them based in the following components: IoT middleware, data repository, data processing, stream processing, cluster management, cloud environment, data access, security, and machine learning. Ref. [35] shows that the FIWARE Platform covers all the main domains expected in Smart City solutions. FIWARE is an open system that offers a public API and is compatible with main communication protocols which has been successfully deployed, for example in some cities in Spain and Argentina. In addition, it is used in some research projects. Reference [37] explains the main components of FIWARE platform and offers a light application to illustrate the usage of the FIWARE platform to build rich applications for smart cities. The FIWARE community [38] envisages an attempt of an official approach to a Smart City solution with the aim of providing added value from the data handled by the context broker. Finally, Ref. [39] does a performance evaluation of FIWARE based on smart cities’ needs. It indicates that the use of FIWARE as a core element can rely on a basic configuration for low load levels, but in the case of high load levels deploying an appropriate shared cluster to be used as global data storage facility is recommended. That work used the official FIWARE agents to access to the data from heterogeneous data sources. In addition, it is worth explaining that the implementation of IoT agents and connectors between data sources and middleware layer need to be improved to support the load of current and future large-scale scenarios for IoT. To improve this flaw, the current work by FIWARE developers is targeting the creation of its own methodology and implementation of the agents to improve the platform performance. The FIWARE platform has been selected to form the device and middleware level component of the architecture presented in this paper and it has been customised to adapt it to Smart Port-City expectations.

### 2.3. Top-Layer Services for Smart Port-Cities Based on IoT

The basis of IoT technologies are heterogeneous, measurable, understandable, and interconnected objects. Their implementation in cities and communities provides applications and services for Smart Cities. Reference [40] provides several definitions of the smart cities concept and an analysis of the evolution of smart cities-related research. Reference [41] identifies the components of a smart city and related aspects of urban life, the key dimensions of a smart city, a list of indicators for smart cities assessment in some rating systems and examples of initiatives. The study in [40] is done from an Information Systems perspective and is focused on number of aspects of smart cities: smart mobility, smart living, smart environment, smart citizens, smart government, and smart architecture, as well as related technologies and concepts. However, not all technological services in cities can be rendered to a “smart city” concept [42]. According to [43], only those that make extensive use of IoT, advanced communication networks and enable user-device interaction can be considered attributable to this concept. The smart city approach has been now widely adopted by cities worldwide as well as via research activities. In [42], several significant examples of top-layer services in smart cities are explained. Spain stands out in smart city real deployments, especially in Barcelona [44] and Santander [45].

Northern-Europe countries have been focused on energy through the AIM [46] and IntUBE [47] projects and on air pollution management through GreenIoT [23]. Internationally, IBM launched a series of deployments creating ad hoc private company units for smart city projects, especially in America [48], while China has embarked on an ambitious initiative throughout the whole country to strengthen city management and to improve city services and functions [49]. Among the previous, the most frequent initiatives are those focused on smart civil infrastructure, air pollution, demographics, smart urban mobility, and smart energy.

There is a smaller volume of literature focusing on the application of top-layer services to smart ports and smart port-cities. The work in [50] explains that, to some extent, the smart port concept can be considered a subset of the smart city. It provides examples of successful implementation of those technologies in relevant ports, indicating that they can improve the port terminal performance, customer satisfaction or reduce the environmental impact. Reference [51] lists a considerable number of smart port initiatives, some smart ports term explanations and a classification of a smart port’s activity domains and subdomains. It consists of four main activity domains: operations, environment, energy, and safety and security, which are, in general, pretty aligned with the smart city concerns exposed in Figure 1b.

There have been some trials to establish a formal classification of top-layer services for smart cities (we have clarified that smart port-cities are a subset of the latter. For the paper, authors decided to use the taxonomy proposed in [52]. This classification was chosen because it enables the combination of smart services dealing with more than one smart city dimension, which is aligned with the hypothesis of our work. For example, persistent high traffic congestion or other transportation/mobility services produce harmful emissions to the atmosphere (NOx, CO_2,_ PMs, etc.) while affecting the local life quality, shaping the perception from the citizenry and potential future civil infrastructure investments and demographic evolution. Some of the former (public safety or demographics) deal with smart living and smart governance. Therefore, a classification system considering multidimensional spots of one smart service was preferred compared to a single tag division. The taxonomy selected drills down every smart city service under three categories, following the so-named “DMS” schema, where “D” stands for data, “M” refers to analytic methods and “S” encodes the service description.

Following that taxonomy, the work in this paper can be properly framed in a “traffic, weather and port-activities” data (D), “co-design and implementation of a composite indicator” (M) and “interpretation of real-time and predicted values” (S).

### 2.4. Real-Time Composite Indicators in Smart Port-Cities

A composite indicator (CI) relies on a mathematical combination of individual indicators that represent different dimensions of a concept. The main objective of this work is to design a system leveraging the IoT data and techniques to represent a complex phenomenon using composite indicators.

Some advantages of using a CI are ease of interpretation, facilitation of rankings, appropriateness for acknowledging evolution through time, provision of big picture and benchmarking suitability. On the other side of the coin, CIs may send misleading messages if not interpreted properly, may entail loss of information, may deviate focus on relevant dimensions and requires high quantity and quality levels of the data to be used to populate all sub-indicators [53,54]. However, if properly tackled and analysed, it can be a paramount keystone for future smart port-city policies, according to the authors.

The usage of CIs has long tradition in some sectors, especially in measuring country performances [55]. However, there is a trend on an increasing use of the composite index methodology—first detailed and explained by the European Commission (EC) in 2002 [20]—for representing interesting indicators in other realms. Some examples were found focusing on land use type and ecosystem services in urban contexts [56], on environmental sustainability (e.g., the Environmental Sustainability Index proposed by the World Economic Forum (WEF) [57] or the Port Environmental Index of the PIXEL project [58]), in the urban economy (e.g., Internal Market Index by DGMARKT [59] or Economic Sentiment Indicator by the EC [60]), on societal aspects (the Human Development Index published by the United Nations [61]) and in science, technology and information advances (e.g., Technology Achievement Index by the United Nations [62] or the ICT index proposed by Fagerberg [63]).

Regarding a technical implementation of a CI analytic software, the main reference found is enclosed in an initiative from the EC-funded Joint Research Centre (JRC) together with the Organisation for Economic Co-operation and Development (OECD) published in the “*Handbook on Constructing Composite Indicators*” [64]. This publication provides a complete methodology and user guide to the construction and use of composite indicators. While there are several types of composite indicators, the handbook focuses only on those which compare and rank country performance. This is out of scope of our research but provides a clear view of the composite indicator topic related with constructing a composite indicator and explaining a toolbox for constructors. Some years a European Commission’s Competence Centre on composite indicators and scoreboards (COIN) was launched the COIN Tool, a MS Excel-based software tool to help develop and analyse composite indicators and scoreboards. It provides guidelines on how to develop methodologies to construct robust composite indicators. As previously, the tool was focused on rankings, but it is a clear example of digitising the process of creating and managing a composite index and its results. It is interesting how they apply the formulas of their methodology using MS Excel formulae and interaction interface. Nevertheless, COIN does not process data in real time or take advantage of IoT platforms.

Narrowing down the research scope, another interesting project targeting composite indicators appeared. The CITYkeys project (EU-funded) [65] is focused on discovering and implementing CIs for smart cities. A huge set of actionable indicators were defined in this project, including cyber-security, local food production, healthy lifestyle, ground floor usage, among others; clearly observing the relevant intervening actors, different examples of deployment and thorough descriptions [66]. However, the calculation was mostly based on Likert evaluations, suing very simplistic scorings (e.g., [67]) and static data coming from reporting or literature databases. Although a performance system was proposed [68], the work in CITYkeys was assumed as incomplete for the purpose of this paper. Another implementation analysed by the authors was the “Dashboard thinking cities” promoted by Telefónica using FIWARE commodities in all the procedure [69]. This option was carefully studied considering that: (i) it uses FIWARE Context Broker (ORION) and (ii) other components that were considered to be included in this paper, (iii) it enables a KPI-powered module explicitly designed to create “Key Performance Indicators” (KPIs) for smart cities and (iv) it allows a multi-dimensional multi-service approach built upon open data and open-source technologies. However, it was discarded as there was not found any relevant real application of the concept, seeming for the moment an interesting proposal pending to be validated in practical cases. Besides, it did not consider including options for tailoring the structure of the CI or interacting with prediction options.

The specific experiment of this paper tackles a traffic congestion use-case. Measuring traffic congestion in a systematic way has proved to be useful. As an example, in [70] a congestion use-case in a real port in China was optimised using a predictive algorithm. In this regard, using composite indicators has already been explored in [71], in which a e composite index—Traffic Congestability Value (TCV)—was defined to monitor the congestion by measuring in real-time number of vehicles and average while driving through specific land areas on the East of India.

Finally, fine-tuning the research looking for validated CIs in smart ports, the most relevant work revealed in the state of the art is [51]. There, a Smart Port Index is proposed to allow ports improve their resilience and sustainability. However, in this development, the joint perspective from the city was not included. Besides, the usability was formulated to inter-port comparison and the KPIs values were extracted over literature references and not supported by an IoT-based system.

The previous analysis confirmed authors’ perception that there is not an ad hoc framework and methodology that both smart ports and smart cities can rely on to build useful composite indicators in real time.

### 2.5. Machine Learning for Traffic Congestion Forecasting

Although not part of the main hypothesis of the work, the authors explored the state of the art of predictive models to forecast traffic and traffic congestion. In particular, this matter is framed within the supervised timeseries modelling, which is a discipline within machine learning.

Few consolidated tools have been found on relation of road traffic prediction for port-cities. The most relevant proprietary solutions are AIMSUN [72], PTV Vissim [73] and IBM traffic prediction tool [74], which has been successful in port-cities in Singapore. Regarding open tools, Veins [75] is the most referenced and used, in a field that stands out for tailored and customised approaches.

About the machine learning models used, there has been an evolution since the “beginning” of this field of study, which date back to 40 years ago [76]. The most used techniques in modern approaches are neural and Bayesian networks, fuzzy and evolutionary techniques, applied to specific application environments with different accuracy outcomes [77].

The expectations on this paper are to replicate some of the already-validated models and methods of the literature and have them applied to a real composite indicator use-case. The objective is not to advance the state of the art on this field, but rather to find a way to merge these technologies with a real-time framework based on IoT concepts.

## 3. Materials and Methods

### 3.1. Framework

The core of this work consisted of the design and implementation of a technological framework to calculate and represent a smart port-city composite indicator (CI). The construction of the software pillars of the framework was driven by the authors taking advantage of the advances of the research project PIXEL [78]. The authors are active participants in that project, which has the aim to develop an IoT-based platform enabling the interoperable interconnection of data sources in ports towards operations and environmental impact optimisation.

The intention of the proposed framework is to properly process raw data to become actionable information in composite index format. To round out the complete system, the framework is escorted by a methodology for its deployment and usage (see Section 3.4).

For the scope of this paper, using the results of previous research [79] has allowed the authors to end up with a solid basis to deploy the framework. Some reference architectures leveraged were RAMI4.0 [80] and IIRA [81], which provide a consolidated layered-approach for IoT service-oriented implementation for practical deployments in a mixture of Industries. The essence was adopted, while specific additions and tailoring were necessary. Drawing from the heart architecture of PIXEL, the framework designed to validate our hypothesis makes use of certain components, altogether with some custom modules and carefully selected extensions. Figure 2 depicts the architecture’s modular composition.

With the vision of delivering an actionable, practical, simplified IoT architecture for enabling composite indicators calculation in smart port-cities, a set of mandatory elements were to be included. First, from the IoT-technical perspective, there was the need to collect and pre-process the data. Afterwards, it needs to be semantically annotated to be interoperable and used by the upper layers. Visualisation and storage are also required (at least, on a bare minimum expression) on every IoT-service system [82]. Second, according to the nature of the service to be provided (see taxonomy in Section 2.3), a series of modules must be embedded; (i) forecasting values of the indicator, (ii) long-term storage for training predictive models and for statistical analysis, (iii) execution of the CI computation and (iv) intelligence to orchestrate the data and process flow. Moreover, the authors made the strategic choice of relying on micro-services and containerization for service provision in order to facilitate development and deployment [83].

All the previous considerations, added to other requirements summarized in Table 1 led the authors to propose the architecture depicted in Figure 2b. Besides, the rest of this section aims at describing succinctly the main function of each module, the technologies selected and the development details for this work.

#### 3.1.1. Data Broker Module

This is a classic IoT component with the crystal-clear mission of gathering into a centric element all the “entities” that provide data to the framework. A crucial concept in this module is the “data format consistency”. Here, the design in this work was to work under uniformed syntactic and semantic structures for all data sources incoming. Hence, an additional processing element must be incorporated (*interoperability agent*) to make raw data comply with the expected schema. Authors have prepared the framework (and encourage all future users) to use standardised interfaces to communicate this module southbound and northbound. Taking advantage of the use of the open-source reference for IoT solutions, all data formats in this experiment have been designed to follow FIWARE NGSI data models [32]. For the implementation of this module, the project PIXEL was the main reference. While the Data Broker module selected technology is FIWARE ORION [34], the interoperability agents are not mandatory to be implemented with a closed specific technology. Recommendation from authors is to use the *pyngsi* Python framework provided by PIXEL project [84]. The selection of ORION brings an IoT flavor to the whole framework, guaranteeing the real-timeness of the system thanks to the publish-subscription approach. Instantiation of this module and agents in the experiment of this paper is explain in Section 3.4.1.

#### 3.1.2. Data Storage Module

The goal of this module is to guarantee persistence of the data feeding the composite indicator. The most significant conditions to be met by this module were high availability, containerisation, open-source, and low-resources consumption. Instead of opting for complex implementations of data storage modules such as PIXEL Information Hub [85], Apache Hive [86] or the classic relational databases like MySQL [87] or MariaDB [88], the choice for this work was Elasticsearch [89]. This selection was preferred over simpler technologies such as MongoDB [90] due to Elasticsearch’s comprehensive data flattening, easy-to-use programming interface and automated filtering capabilities.

#### 3.1.3. Orchestration Module

This acts as the orchestrator providing the intelligence to the procedure. As commented, one of the decisions was to deliver the backend modules (CI calculation and prediction) as “independent” containerised services. This module is in charge of coordinating the process to rhythmically execute those services, indicating which data must them be fed with and other variables coming from user preferences. Some options were discussed for the implementation, such as Node-RED [91], Docker Swarm [92], Kubernetes [93] or PIXEL’s Operational Tools [94]. However, as the application for Smart Port-City composite indicators proved to be straightforward from a service layer point of view, the authors decided to design this module in the framework as a combination of Docker compose [95] instructions and Linux bash scripts scheduled using *cron*.

#### 3.1.4. Visualization Module

The options for visualizing composite indicators are motley. Ranging from direct-database-connection tools such as Kibana [96] or Grafana [97] for simple inspection to a fully customized layout, the state of the art is plenty of opportunities. Drawing from the requirements in Table 1, the election had to be customizable enough to allow the authors build their own visualizations and configuration forms. Therefore, based on the baseline approach tackled by PIXEL project, the framework Vue.js (JavaScript) [98], was selected. On that regard, the authors developed a set of tailored tabs with dynamic graphs to represent the information associated to a CI utility. Details on the application to this experiment can be found in Section 3.4.4.

#### 3.1.5. Dual Computing Approach

While the majority of the previous modules are functional enablers, carrying out mandatory tasks for the correct execution of the service, the core calculations to extract valuable knowledge out of the data are still to be revealed. Authors have designed the framework to have two clearly differentiated “data processing spots”, as it can be seen in Figure 3. In a simplified way, raw data comes from the left block in, it is converted into a common agreed format through the interoperability agents, it is stored and then it is retrieved from the CI mathematical backend calculations to obtain the final single metric.

This structure was decided by authors to allow flexibility to change/correct/adapt/add further data sources in a constrained part of the architecture, without affecting the structure of the whole framework, leaving the calculation from input data to the composite indicator (and subindices) as a fixed code not supposed to be changing throughout the time. Additionally, this way, the most burdening computation could be jointly borne by different equipment.

(1) Data processing spot #1: KPIs computation block

Following the rationale behind one of the tools created in PIXEL, the Port Environmental Index (PEI) [99], the authors decided to address the CI calculation from a cascade down-top indicator flow. Henceforth, for this work, it was adopted the approach of considering each piece of data input as one KPI (Key Performance Indicator) to be monitored, constituting one leaf piece of the indicators tree.

This block constitutes the basis of the CI calculation, aiming at converting data into actionable KPIs and make them ready to use is what the agents perform. NGSI prefix is used to explicitly state FIWARE compliance. It is worth to mention that the NGSI agents’ development is an action that will deviate from one “Smart Port-City composite index” use-case to another. As data will be different, the treatment at this level will vary and it will be needed a careful, tailored analysis to handle the development of the agents. The functions that all NGSI agents must provide are the following:To retrieve the data from the original source: the framework has been designed to accept a two-fold connection mode: (a) the agent actively queries the data source origin following a periodic pattern. This option will apply whenever the data is behind a reachable API; or (b) the agent includes an embedded data broker so that the active origin can publish on it. This case is usually present in the cases where built-in IoT stations or smart sensors are used.To process the data and convert it to KPIs: this development will be different for each data source, becoming the most craftsmanship development when a composite indicator tool wishes to be deployed using our framework. It might range from a simple format conversion to a complex data relation, combination, and construction. Additionally, each NGSI agent may have different number of inputs and outputs. Despite the fact that the usual case (see Section 3.4.1) is to realise a 1:1 relation, the framework has been prepared to accept 1:N, N:1 and N:N setups as well.To update the entity in the Data Broker: The main goal of the agent is to make data reach the information layer of the IoT stack of our framework. Hence, the KPIs obtained are submitted to the Data Broker (ORION) via a PUT HTTP—Hyper Text Transfer Protocol—message in order to update the “KPI entity”. The format selected has been to extend one of the FIWARE Data Models: *KeyPerformanceIndicator* [100]. More details can be found in Appendix A.

(2) Data processing spot #2: Composite Index calculation and prediction block

This block consists of a series of calculations that will be invariant, following a clear algorithm. These calculations are transparent for the actors intervening in the composite index definition and usage, remaining as a “black box” that takes some inputs, processes them, and provides an output (result of the execution). According to the designed framework methodology (see Figure 4), this processing spot performs a non-serialised 6-step mathematical operation procedure:
(1)To configure the CI inputs, the structure of the tree and the associated parameters, such as aggregation methods, normalization methods and weighting values through a developed visual interface. See Section 3.4.4. to discover the utilization of this component.(2)According to the parameter set in (1), including the scheduling, the periodic calculation of the index. First, the KPI values are properly normalised (crucial step on a CI procedure). Then, starting by the leaf node, a cascading algorithm including aggregation, weighting and combination leads towards a final index value. Technologically, this component has been developed as a dockerised (packaged in a Docker container [101]) standalone Java application.(3)To configure the predictive component of the framework by the user, specifying batch sizes, periodicities and model to be used for the inference of KPIs.(4)According to (3), the training module groups the KPIs stored, fits again the pre-trained model selected, updates it, and makes it ready to be used. Technically, this module has been developed as a dockerised Python script.(5)To apply the prediction over predicted KPIs. This module has also been developed as dockerised Python script.(6)To visualise the results of both real-time calculation of the composite index (2) and the observation of the predicted evolution of the composite indicator via a specific dynamic graph (5).

### 3.2. Proposed Use Case

The experiment conducted in this paper consist of envisioning, designing, implementing, and testing an IoT-powered composite indicator reflecting the traffic congestion in the interface between the port and the city of Thessaloniki in Greece.

The Port of Thessaloniki is spotted very close to the city center, which has been throughout the years built around the economic opportunities that a port brings. During the last ten years, a huge number of diverse activities have been transferred to the surrounding area of the port, which has been increasingly selected as the ideal place to establish office-based businesses. Several luxury hotels and new business centers, including brand-new and re-modelled buildings have been constructed just outside the port area, thus being directly influenced by normal daily port activities. Establishing a physical base near to the port may have certain disadvantages, such as the noise or the (possible) impact of dust in the air. However, there are also pluses, such as the views offered to customers and employers and the privilege of direct contact with the business heart of the city.

For what directly concerns this work, in the selected use-case setup (Figure 5), the aforementioned port activities and growing businesses are hugely impacting the traffic congestion in the surrounding streets to the port and city center. Moreover, the industrial area of Thessaloniki, the main bus station and logistic centers are all located at the west side of the city. The main truck gate of the Port is on its west side resulting in a significant amount of traffic during the peak hours. Here, peak hour is determined by the truck movements (entries and exits of the port), which curve is similar to Figure A4a) (in Appendix B). The context explains what makes the area already environmentally burdened.

The objective of the use-case is to demonstrate the hypothesis proposed by using the framework outlined in Section 3.1 towards allowing a port-city to build policies upon a legit, reliable, real-time based composite indicator.

According to the previous, the event that is wished to be modelled is the traffic congestion at the interface of the port with the city. The composite index defined is, therefore, a *Traffic Congestion Index (TCI).* The TCI must consider the different elements intervening (in a per-day assumption), as well as the different actors that are involved in the policies definition.

On the one hand, the traffic congestion level is represented by the number of vehicles at the gates of the port (real port-city interface). However, there are other factors influencing the congestion level, such as port activities (e.g., by the number of ships being operated), the weather conditions (e.g., by affecting port operations decreasing productivity [102] or noticing that rainfall leads to more traffic congestion in cities [103]), the seasonality (season, month, day within the week) and, naturally, the contribution for the city itself, i.e., traffic on the port surroundings attributable to the city. Higher number and of more complex nature requirements could be argued to be considered [104], but for the scope of this work, the main factors affecting the congestion were the previous.

On the other hand, the main actors involved in the traffic congestion event to tackle with the TCI are the port authority, the city municipality and, indirectly, the citizens of Thessaloniki. During the work exposed in this paper, an active participation was received from the port authority, funneled via the participation in the PIXEL project. The Port Authority of Thessaloniki (THPA) collaborated within that project by providing traffic data at the gates of the port, by co-designing the concept of the experiment (possible influences in the traffic, relevant data sources to explore), by indicating a series of functional requirements and by outlining the visualization layout that a port authority would like to have in that kind of experiment. With regards to public participation, the authors did not have close enough contact with the local community nor resources to involve, in this case, citizens, the municipality or other entities into the experiment. As this use-case is aimed to serve as a starting point—proof of concept—, this was finally deemed not mandatory thus not included in the methodology of the experiment. However, to bring their perspective into the experiment, authors designed the system flexible enough to allow the introduction of new parameters at any moment. Additionally, this intervention has been also properly considered within the methodology (see steps 1 and 2 of the methodology in Section 3.4).

The experiment draws from historic and real-time data (see Section 3.3), deploying a number of IoT techniques (see Section 3.4) leveraging the framework developed (see Section 3.1), ending with the visualization (see Section 4) and interpretation of results (see Section 5). The paper does not aim to go beyond in the effective implementation of policies in the city of Thessaloniki. However, according to the authors, the work carried out might clearly be used for that purpose.

For replication of a smart port-city composite index experiment, and for clarifying the use-case composition, Table 2 illustrates the main features and characteristics. The framework exposed might cover a wide range of use-cases, which could be represented using the proposed format.

### 3.3. Data

The experiment has been run with a series of raw data coming from diverse sources. In this section, we aim at describing the historic and real-time data used, as well as the pre-processing made to store and utilise it properly.

#### 3.3.1. Traffic at the Gates of the Port Using Radio-Frequency Identification—RFID—Sensors

The baseline data for the experiment comes from two RFID sensors placed at the gates of the port of Thessaloniki, as depicted in Figure 6. The gates are 10A and 16 (as numbered by the port) and are equipped with RFID-based sensors allowing them to identify and catalogue all vehicles entering and exiting the port.

The data has been provided to the authors of the paper under the collaboration with the Thessaloniki Port Authority, S.A. (THPA) framed within the PIXEL research project. This data has been granted by THPA and served by its IT department via a web API created explicitly for the mentioned project. Internally, the data is managed by a Port Community System.

Historical data from April 2018 has been used. The experiment was decided to be kept up to end of February 2020 in order not to have data influenced by the COVID-19 pandemic. The “raw” format of the data is represented in Figure 7.

Therefore, for using the data for the TCI, a pre-processing step was needed. Data from April 2018 was queried from the past entries/exits of vehicles and the same structure was established for real-time and future continuous usage of the composite index. The process consisted of retrieving all vehicles in the period, filtering by the *“gate*” field, analysing the timeframes they crossed through the gates (fields: “*time*”), establish hourly ranges and count afterwards the number of vehicles on that range. This exercise was materialised in Python scripts and the data digested was store in a proper database to later feed the model training according to the software design of the framework (see Section 3.1).

#### 3.3.2. Traffic at the City Provided as Open Data

For including city traffic data from the surroundings of the port, the authors used an open data source consisting of digested information from car fleet equipped with Global Positioning System—GPS—providing location and speed information of the city of Thessaloniki. This data (with especial interest due to the historic offering) was served by courtesy of CERTH-HIT (member of PIXEL) via the TrafficThess website [105]. This website has varied information of the different roads of the city (referred as links) updated each 15 min, which is perfectly aligned with the used baseline dataset for this task. The pre-processing followed over the raw data involved: (i) selecting five of the surrounding links to the gates of THPA, (ii) extracting the data for those five links from April 2018 till February 2020 (same reasoning than before), (iii) selecting the interesting fields of the information provided, (iv) building a Comma-Separated Value (CSV) table with the average of the average speeds of the 5 surrounding links. This is briefly illustrated through the images in Figure 8:

As explained, TrafficThess’ historical is sampled every 15 min. To make the calculations consistent, both historic and real-time data was averaged and condensed to have a 60-min granularity. Thereafter, the dataset was properly stored, ready to be used for TCI computation.

#### 3.3.3. Vessels Berthed in the Port

Similar to the traffic at the gates of the port, the authors were provided with historic and real-time data of the vessels processed in the Port of Thessaloniki courtesy of the Port Authority. This data was served by the Information Technology department of the Port of Thessaloniki via a REST API web analogous to the previously mentioned. The query to that API returns all the vessels that were operated (one API per year) in a JavaScript Object Notation (JSON) format including rich information and details of every single vessel in a year. This information is only timestamp-referred by including fields of “*start_work*” and “*end_work*” of each ship, therefore certain data pre-processing after acquisition was needed. The procedure followed was: (i) downloading the data of all vessels processed in April 2018-February 2020, (ii) fine-tuning the timeframe, (iii) grouping, filtering and counting the vessels to prepare a CSV file the proper info: number of vessels at berth/manoeuvring in the port separated by periods of 60 min This is briefly illustrated through the images in Figure 9:

Regarding the pre-processing, most data from the raw retrieval was ignored except for counting the number of vessels in a timeframe. Filtering by the fields of “*start*” and “*end*” work, the dataset was built. The technology used for the pre-processing were Python scripts, taking advantage of the features of the library *pandas*.

#### 3.3.4. Weather

The authors have used the free web service provided by Stratus Meteo of Greece [106]. Different sensors are installed throughout Greece and for this case we made use of the one closest to the Port of Thessaloniki (presented in Figure 10).

Historical and real-time measurements of temperature, wind speed, precipitation intensity, humidity and dew point (among others) were ready to use considering the following hindrances: the information retrieved from this external service was only served with a daily granularity. In that regard, a preliminary action needed was to re-shape the data to adjust the structure for the posterior modelling. Moreover, the dataset of weather was also available from September 2018, therefore diminishing the quantity of data for training and validating the model and the expected accuracy.

Table 3 shows a summary of the data that has been used for the experiment:

### 3.4. Methodology of the Experiment

The experiment has consisted of deploying the framework of Section 3.1 for the use-case outlined in Section 3.2. To conduct the experiment, a methodology of various steps was envisaged and took place. The explanations below aim as well at being a guide-for-using for any person wishing to replicate a similar practical case using our framework.

On one hand, the “conceptual” methodology was comprised of the actions needed to be tackled by the involved people (port, city, citizens, societal agents). The idea is to have a useful CI that will lead relevant actors to set sustainable, smart policies upon it. Therefore, this activity is of paramount importance. For this experiment, the only actor that participated was the Port Authority of the Port of Thessaloniki, under the scope of the project PIXEL. The “conceptual methodology” has three steps, whose details are shown in Figure 11. The purple boxes represent the moments where the technical staff will need to intervene to guarantee effective deployment of the framework.

On the other hand, the framework had to be customized and instantiated for conducting the experiment. Apart from enough equipment material to run the modules integrated (see Section 3.4.4), the technical intervention went as follows:
Definition and development of the NGSI agents: developed by technical experts after gaining information of the data available (see Section 3.2).Definition of the composite index (TCI) by stakeholders using the visual interfaces developed in the framework.Setup and running of the predictive component, from user parameters to actual software running.Configuration, implementation, and integration of all the pieces together to have a real-time composite index calculation.

In the next sub-sections, we are describing the materialization of the steps above for the experiment developed in this paper.

#### 3.4.1. Definition and Development of NGSI Agents for the TCI

The technology selected to implement the NGSI agents was the Python *pyngsi* framework. It was developed as part of the H2020 PIXEL project. Writing a NGSI agent using this framework avoids developing them from scratch, because it offers a clean code structure and a documented tutorial. Therefore, the developer of an agent can focus on writing his own logic to build NGSI entities.

Figure 12 shows how the agents developed enable to process experiment data from heterogeneous data sources, content formats and protocols using a common interface, convert the data received in the NGSI entities based on FIWARE data models and send it to the Orion Context Broker. Specifically, for the use case, the data sources were two-fold: historic datasets consisted of structured CSV files while real-time data was to be retrieved by specific HTTP requests to REST APIs. This drove the authors to create two agents per data source: one for the historic and another for the real-time connection. In both, data was transformed in a JSON schema, covering the key-value representation of NGSI v2 context data. As designed in the framework (Section 3.1), the NGSI v2 data model defined is a custom extension of the *KeyPerformanceIndicator* FIWARE Data Model. It was extended following the design principles and guidelines offered by FIWARE official documentation [107].

According to the previous, all NGSI agents needed to be developed in one experiment must follow the three-step processing. Within the syntactic and semantic transformation, several other computations such as filtering, grouping, cleaning, discarding may take place. Additionally, all interoperability agents have certain inseparable parameters, which are the refresh time (RT) and the “*kpiName*” field name. In the experiment of this paper, a total of seven agents were developed to ensure proper storage and usage of historic and real-time data. In Table 4 details are found of their development.

#### 3.4.2. Definition of the Traffic Congestion Index

According to the use-case definition, the TCI composite index represents the congestion status due to diverse factors. To build the CI structure, the authors proceeded to analyse the data, establish a levelled tree layout and weighted the nodes. This exercise must be tackled jointly by technical staff and stakeholders in a potential replication of the experiment.
*Grouping of data into KPIs*: Following the state of the art analysed in the projects PIXEL and CITYkeys, the authors decided to group the data for feeding KPIs by common origin and meaning [99]. This drove the design to couple the traffic on one side (gates of the port and city), vessel information on other side and weather information on a third and final string. The leaf nodes (KPIs) were also individually separated by isolated pieces of information (Gate 16 of the port, Gate 10A of the port, vessel count, temperature, wind speed and precipitation intensity). This led to a three-levels composition, being the KPIs the leaf nodes, three subindices and the TCI as the root node, resulting in a 7:3:1 matrix.*Exploratory Data Analysis (EDA) of available historical data*: Authors carried out a thorough EDA of the historic of data (see Section 3.3). A summary of that EDA is attached in Appendix B. The main aim of this EDA was to discover how the data performs through time, noticing seasonality, and, mainly, to find the correlation between the different data with the main reference source: the traffic at the gates of the port. The results of this correlation were the following:*Weighting*: The last-but-one configuration for the CI was to select the weighting method and values to give to each node. This is a crucial action that has been widely studied in the composition of CIs [108]. There is not a universal weighting method and it must be analysed case by case, introducing a challenging choice [109]. The most used method is equal weighting, followed by analytical methods (regression analysis, benefit-of-doubt, principal component analysis), with opinion-based methods only used marginally. The choice depends on the nature of data and indicator items: (i) equal weighting tends to be used when no historical data is known, (ii) analytical methods are case-to-case analysed depending on the items and (iii) opinion-based methods are mostly used in social sciences (cases where the target indicator is highly subjective). In this experiment, the authors opted for an analytical method driven by the analysis of historic data available. The specific scheme selected was to base the weighting values on the correlation of all items with the main reference item. For establishing those values, g the authors came to map KPI-weight, sub-index-weight, which brought to solve three equations systems (see numbers and structure of the points above).
(1)∑Wi×Corri−other_sources_in_string=1

The relation among KPIs is known by Table 5, therefore the Gauss method was used to solve them and come up with the weights, which are displayed in Figure 13.
*Aggregation method*: According to the literature, there are three widely used aggregation methods [108]: additive aggregation, geometric aggregation and non-compensatory aggregation, being the first one the most popular by far. Additive aggregation provides transparency and allows a simple understanding of the results, being much dependent on the synergies between items. Geometric aggregation also provides a good understanding of results but requires uniformity in measurement units and scales, being very dependent of synergies as well. Non-compensatory methods are fit for cases where the indicator is going to be deployed in several instances (e.g., ports, countries) and those aim to be compared and ranked. Non-compensatory methods are the most computationally expensive of the three. For this experiment, the authors considered that the relevance of all indicators was not equal, being the traffic at the gates the reference item. Noticing the previous, the choice was to select additive aggregation. However, the framework developed in this paper has been designed to allow the selection of any of them for future uses.*Normalization method:* The items to build the indicator must be comprehended in the same scale in order to be aggregated. Thus, a normalization step was included in this computation. A lot of methods for normalization are available, and to be able to finally select one as definitive several robustness test must be done. The objective of the work in this paper was to demonstrate the use of these calculations using a specific IoT architecture rather than going deep into normalisation arguments. Scale method was discarded due to magnitude variation, as well as the ranking due to its only execution in Thessaloniki. Z-score method and Min-Max were the main candidates and, considering that in this experiment the authors had valuable historic data for more than 18 months, the Min-Max option was selected.

#### 3.4.3. Predictive Component: Training and Validation

In the scope of the composite indicators using the proposed framework, the prediction will always drill down as a timeseries forecasting problem. According to the revision outlined in Section 2.4, in this case, a model must be trained with past data and requested to infer new values for a certain horizon using the same period than the available dataset, so a custom predictive model had to be used by transforming timeseries problem into a classical structured supervised ML problem. In those, no fresh data is required as input to a “prediction black box” to influence the prediction. Therefore, what must be done is re-train the model periodically to be sure to be exploiting the new data as much as possible.

For this experiment (see Section 3.3), the periodicity of data is 60 min (except for weather, that is daily). Thus, it was decided by the authors to establish 60 min as the periodicity and one day as the horizon.

Regarding the selection of the model, several options were available (see Section 2.4). Especially interesting were Autoregressive Integrated Moving Average—ARIMA—models or time series regression models such us Seasonal ARIMA (SARIMA) or Autoregression Vector [110], among others, widely available in Python libraries in the open-source community. Authors realized, thanks to the EDA, that some data registers presented missing values. Besides, both traffic in the city and weather data presented strong daily and monthly seasonality. Henceforth, a model engine needed to be selected that properly handled those criteria. According to the authors, the most appropriate selection was to use Prophet [111], a general purpose timeseries forecasting tool developed by Facebook. This election was strengthened while observing some of its traits: easily integrable in agile deployments with heterogeneous data. Detailed analysis of the use of other state-of-the-art models such as gradient methods were applied in one activity of PIXEL project [112], which outperformed Facebook Prophet by significant margins, but required more effort and data to be integrated, hindering the automation capacity and seeming overkill targets for what is needed in a Smart Port-City CI application.

Regarding re-training and evaluation, the framework has been designed to be re-trained under request. At each re-training iteration, the model will be evaluated on left-out datasets following a 70/30 dataset split, using different error metrics. For this experiment, the model was re-trained three times, resulting in a validation that can be consulted in the repository of the materials of the paper:(1)To convert and clean the historic CSVs into the accepted data format for training the Prophet model. Here, the attributes had to be adapted and seven different models were created. The framework and methodology designed (see Section 3.1) established that all KPIs must be predicted, and then the CI is calculated after those values. For that reason, one model per each KPI was trained and used. This was developed by the authors using Jupyter Notebooks [113]. The procedure was studied to be replicated in an automated way for (2).(2)To create Python scripts that gather and group the KPIs data from the data storage and convert, clean, and adapt the information for re-training the seven models. In this experiment, three re-trainings took place, according to the framework usability evaluation depicted at the beginning of Section 4. The outcomes were new models that have used more historic data to be trained, therefore more accurate and usable. For storing and retrieving those models (binary files), the Python built-in library *pickle* has been used.(3)To apply the models. As commented, the forecasting horizon was set to 1 day with a 60 min periodicity. This way, each morning, the framework runs the inference over the trained models. Prophet models do not need input data; therefore, a prediction is requested (automated through a Python script) with that periodicity and horizon. The outcome (future timestamps with predicted TCI values) are used to be represented for the user via the UI component.

#### 3.4.4. Component Integration and Deployment

When all modules were developed and ready to be used from an isolated perspective, then the authors proceeded to the integration and deployment. Using the authors’ university department own premises, a prototype of the framework was deployed to run the use-case of the experiment. The different modules (as designed in Section 3.1) were integrated, connected using the proper ports and following the containerization (Docker) approach, resulting on a continuous working system. The schema of Figure 14 aims at representing the module interactions, data flow and “network configuration” of the deployment of the framework for this case.

As it can be seen, the deployment follows the design of Figure 2 (Section 3.1), keeping consistency with horizontal and vertical competences of every module. However, it is worth to mention that some additional elements—not explicitly stated in the framework design—were needed to achieve effective integration. These additional elements, and a summary of the integration flow, are narrated below.

The NGSI Agents retrieved the raw data from the different origins, converted them into the proper format based in the FIWARE KPI datamodel, and then updated the attributes of the related FIWARE ORION entity. As one mandatory addition, the Data Broker modules had an associated MongoDB database. This database is embedded natively with ORION in order to persist the last value of the entities’ attributes, which is named context data. Afterwards, to be able to connect the Data Broker with the long-term storage database of the framework (Elasticsearch), another auxiliary element was required: FIWARE Cygnus [114]. This component subscribed to the ORION entities and created a sink to connect with the upper-layer database. With that purpose, the *NGSIElasticsearchSink* [115] was used.

The orchestrator module was deployed as a crucial piece of the framework. The integration consisted of properly using Linux *cron job* to schedule the prediction scripts and TCI calculation.

Atop those, the Elasticsearch REST API module—another additional element—was needed to act as a bridge between the data storage module and the visualization UI. It was built using NodeJS [116] and the Elasticsearch Node.js client [117], with the purpose of avoiding queries to Elasticsearch directly from the UI. This module prepares the data with the specific format used by the map and the different charts that are displayed in the dashboard section of the UI and stores the tree specification in the database. Finally, the UI was served like a static web page by an Apache Server [118], so it was able to be accessed by the users via a common web browser. This web browser makes the necessary requests to the Elasticsearch API module.

Regarding technical details of the deployment, all the modules were deployed in the same machine, except the data source REST APIs (city traffic, weather and THPA). This machine belongs to the authors’ investigation group private cloud and its hardware specifications are detailed in the Table 6. In addition, both the software specifications of the pre-built modules (FIWARE components and Apache Server) and of the technologies used to develop and run the other modules are detailed in the Table 7.

#### 3.4.5. User Configuration

Once the modules are integrated and the deployment is set, the last step of the methodology was to adjust the visualization and other configurations to interpret the CI. These configurations—in a supposed replication of the experiment – must be done by the actors interested in the composite index. In this experiment, the authors took the liberty to configure the solution assuming how it might be used by such actors. There were two configurations, and all was enabled to be set via user interface:

(1). To translate the CI calculation configurations for actionable instructions

The TCI definition is explained in Section 3.4.2. The CI structure and weights were there defined from an operational point of view. The configuration at this step meant to translate that definition into software instructions. There, the authors created the number of levels of the tree and then, create all the tree nodes, beginning from the root node and finalizing in the leaf nodes. For each node, it was introduced the name (in leaf nodes, it must coincide with the name of a valid KPI stored previously in the Elasticsearch database), description and, depending of the node level, the weight, weighting method, aggregation method and NGSI Agent. Furthermore, it was necessary to select a common normalization method and to introduce the URL of the normalization API, needed to run the TCI (Figure 15). The selected normalization method for the experiment was Min-Max [119] per KPI.

(2). To establish thresholds, reference numbers in the visualization of results

Another section of the UI where the authors stated configuration parameters was the dashboard, that will be explained in detail in the Section 4. The authors inserted the threshold values for low and high traffic in the line graph and the threshold values of traffic at the gates that will be translated to the gates’ color in the map (Figure 16). For the experiment, the configuration values selected were:0 ≤ TCI < 0.2, *Low congestion*
0.2 ≤ TCI < 0.46, *Medium congestion*0.46 ≤ TCI < 1, *High congestion*(2)

## 4. Results

The objective of the proposed framework was to provide usable, reliable, actionable knowledge via a composite indicator. The experiment deployment presented above was aimed at demonstrating the usefulness of the framework. Therefore, results were needed to validate that objective.

In the context of the TCI, the traits that had to be analysed were:(i)composite indicator makes global sense and represents the reality,(ii)predictions are valid and realistic, so that the model can be trusted, and(iii)the interface usability makes it easy to interpret the information to build Smart Port-City policies upon.

As explained in Section 3.3, the historical data in the experiment covers from April 2018 to February 2020 (except for the weather, which is available from September 2018). From that point on (February 2020), this framework was designed, and the data from that moment was gathered in real-time. The Port and the City of Thessaloniki have certain seasonality patterns on the traffic and weather, therefore, in order to validate the framework, the strategic choice was made to select certain dates to obtain results to analyse. Drawing from the EDA carried out, it was noticed that summer months’ performance varied in comparison to winter months; diverse variation within the week and the hour of the day were also observed. Henceforth, it was decided to define 9 scenarios to cover most of the situations that the port-city traffic might experience. Specifically, three timeframes were selected (one week in March, another in August and another in January), selecting three different days at three different hours. The dates selected were Tuesday at 9 a.m., to represent a busy period with regards to vessel operations and traffic at the gates, Thursday at 3 p.m. to reflect moderate traffic period (at average) and Saturday to reflect low port activity and high city life. This way, authors aimed at having enough base to reassure whether the framework will be valid.

For visualising the results, the authors created a UI dashboard, as contemplated in the top-layer component of the framework. It is shown in Figure 17, representing the results of scenario 1.1.

The first quadrant is titled “Current traffic congestion at port gates” and it is formed of an OpenstreetMap [120], which has been created using the library Leaflet [121], including two markers that correspond with the location of the two gates (10A and 16, from left to right) that are analysed. The markers’ color indicates the congestion level in its corresponding gate. At its right, the gauge chart shows the current value of the TCI index. The third quadrant is titled “Historic and prediction of TCI index value”. This line graph shows the evolution of the TCI in the whole day and it is split into two parts: the first one represents real values until the current time while the second one represents predicted values. Furthermore, the graph includes two lines that indicate the upper (red color and TCI value of 0.45) and lower (green color and TCI value of 0.2) thresholds. Finally, the bullet chart represents the current value of the three indices (traffic, vessels, and weather) obtained during the TCI calculation process. Most quadrants’ interfaces of this visualization UI have been created using the library amCharts [122].

According to the authors, the previous screen enables the port and city stakeholders to interpret the current and the short-term predicted evolution of the traffic congestion. Stakeholders will be able, then, with this framework, to realise the real-timeness of the indicator, being able to make timely comparisons and knowing, with a quick glance, the status of the congestion in the port-city interface. The objective (iii) was met.

The same exercise was repeated for the nine mentioned scenarios, obtaining, for each, the value of the TCI on that moment and the predicted TCI values (thus its expected evolution) for the rest of that day (find all the screenshots in Appendix C). In the table below, there is a summary of the results and main conclusions extracted upon them. The rationale behind the column “*Explanation and evolution*” comes from the EDA. In general, the TIC values (and short-term prediction) are considered valid if it can be established a direct reasoning from the conclusions extracted during the EDA.

Observing the results in Table 8 and Figure 18, the objective (i) can be considered achieved. The value of the index is always comprehended in the 0–1 range, indicating 1 as maximum traffic congestion and 0 minimum traffic congestion. In general, values are higher on labor days and lower on weekends, experiencing more traffic congestion in the mornings. The graph below aims at representing those values.

With regards to the last objective, (ii), the “explanation and evolution” column in Table 8 details how the prediction of the TCI meets the logical expected curve. Besides, the performance metrics obtained of the model utilised show little error (Mean Absolute Error—MAE), especially in the most significant data references: traffic at port gates and traffic at port city.

As a point of discussion and utility of the framework, the technical solution developed provides an extension component for PIXEL platform, a potential enabler of FIWARE stack [123] and could be offered as an independent software tool atop those platforms.

Those facts together make the authors conclude, that, on the light of the results, the TCI experiment demonstrates that the proposed framework and methodology might help tackling current and future Smart Port-City challenges.

## 5. Discussion

The authors believe that the outcomes of this work might hugely contribute to cities and ports by implementing an IoT-based solution, which is the main focus of this special issue. Specifically, the TCI experiment carried out may support the effective implementation of policies in the city of Thessaloniki. To mention one, the TCI can be used in the reports of the port and the city to evaluate how the different urban planning initiatives or port land construction works are affecting the traffic in the surroundings. Another port-city policy that could be fed with the TCI are the citizens’ safety associated to noise and pollution levels attributable to that port-city interface. From another perspective, the TCI value could be incorporated to the traffic lights cycle, allowing thus to reduce the congestion which might be a policy objective.

For a port authority, a deployment following the line of this TCI experiment can bring several benefits: first, having a unique point combining a series of relevant available resources, after its collection, having it monitored and published; second, being able (through data analysis) to identify periods of time in which the traffic is above accepted levels, focusing on the areas or procedures that can be optimized, correlating between processes and corresponding equipment, giving priority to areas (or procedures) with realistic reduction potential and thus optimising inbound and outbound truck flows and equipment movements; third, to establish internal process-centric practices, such as gate policies to specify a time slot during which trucks are allowed to enter the port area or relationship-centric practices, such as training operators (of machinery, vehicles, etc.) with the goal of minimising bad practices leading to higher traffic congestion that can have a negative environmental impact. Last but not least, visibility and acceptance from the community. From the municipality perspective, the TCI can be useful for another set of targets. Understanding the main contributor to the traffic congestion may help the administration to apply tailored control, influencing demographics, urban development, taxes or fines. Public bodies may also benefit from experiments like the TCI as a tool for auditing performance and ensuring to keep traffic levels among reasonable thresholds. Additionally, the semaphores and other signaling rhythms adjusting systems would have another input to be fed from. Finally, as a joint benefit, linking with the hypothesis proposed in this paper, the TCI may serve as a meeting point for all the actors above to know the current status of traffic congestion and to agree upon policies.

## 6. Conclusions and Future Research Lines

In this research, the authors have analysed the related work on the fields of smart port and smart city services, looking for finding a software-based solution to address relevant challenges of the smart port-cities of the future. The work was devoted to developing a framework to leverage IoT techniques focusing on the calculation and prediction of indicators representing real-world phenomena. In particular, as an outcome of this paper, the solution has been applied to real data on the port-city interface of Thessaloniki in Greece.

IoT open-source components have been used in the design and deployment of the framework, introducing elements for enabling short-term prediction, high usability and configuration capabilities for the user and an interpretation interface. Heterogeneous data sources have been handled using specific defined data formats, achieving syntactic and semantic interoperability. The software has been successfully integrated in virtualised servers hosted in one machine by the authors. Results were satisfactory in all the scenarios defined for the framework validation.

According to the authors, this field of work will open a wide variety of options to be explored, both from the application point of view (different actors, usability, standardization, methodology) and from the technical perspective.

On the one hand, the co-design and flexibility of the system will allow incorporating data and wills from different actors. For the citizens, having access to such an indicator would mean a step forward in the democratisation of the information and procedures, which can increase the acceptance of the government and of the port activities. This concept aims at boosting the so-called “enlightened political participation” [124] of the citizens to affect the decisions on their environment beyond the periodic voting. This is clearly a field to explore further as the outcome of this paper will offer the citizens and other entities the possibility to access and actively participate in such actions.

On the other hand, additional options for technology research are motley. Specifically, authors would be interested on see this framework as a basis for including more complex prediction models, a higher number of data sources and with more refined orchestration approaches. Regarding the framework composition, the authors believe that the dual approach designed will open interesting research lines; there is a trend in the distributed systems to move computation to the “edge” of the network to bring benefits such as reducing latency, optimizing bandwidth use, improving privacy and security, and alleviating network congestion and traffic in general [125]. In future applications, heavier training and predictions will be moved to the “NGSI agent side” of the system, allowing the central server to process, offer and show more data sources with more accuracy degree closer to real real-time. The fact of being built upon open source consolidated technologies makes the proposed tool a perfect testbed to be leveraged by the community, going beyond classical private customized approaches.

## Figures and Tables

**Figure 1 sensors-20-04131-f001:**
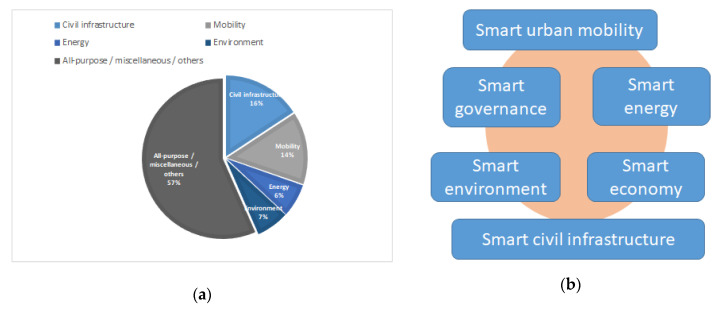
Smart Cities services analyses 2010–2018. Data extracted from the study done in [42]: (**a**) percentage of appearance in papers; (**b**) main areas of interest for deploying Smart City services.

**Figure 2 sensors-20-04131-f002:**
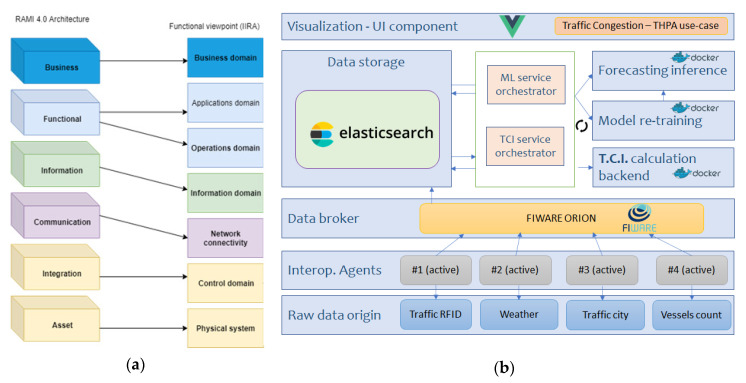
Architecture of the proposed framework: (**a**) IoT layered reference architecture RAMI4.0 and IIRA—image extracted from [79]; (**b**) Modules and technologies of the framework. The figures are aligned to reflect the mapping reference layer - component.

**Figure 3 sensors-20-04131-f003:**
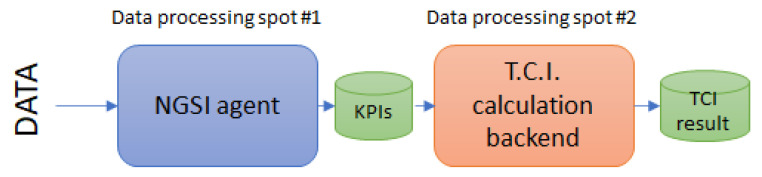
Dual computing approach followed for the proposed framework.

**Figure 4 sensors-20-04131-f004:**
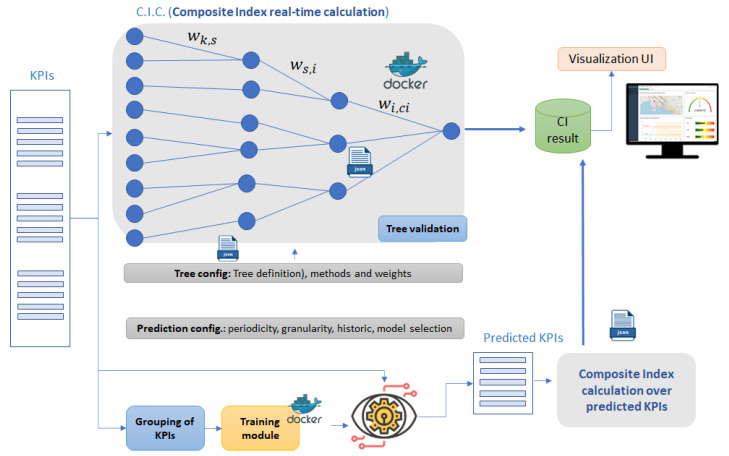
Composite Index calculation and prediction block.

**Figure 5 sensors-20-04131-f005:**
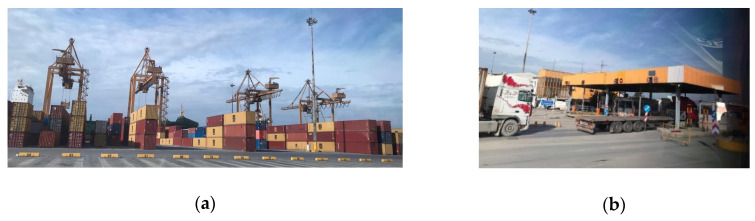
Setup of the use-case. Port-City of Thessaloniki and its interface. (**a**) Picture of cranes of the ports’ container terminal; (**b**) One of the truck gates of the port.

**Figure 6 sensors-20-04131-f006:**
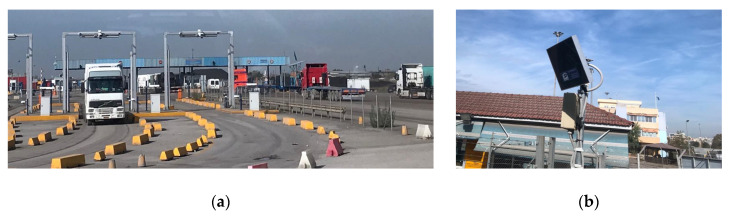
Data coming from RFID sensors in the gates of the Port of Thessaloniki: (**a**) RFID arcs at the entrance to the port of the trucks gate (16); (**b**) RFID tag and vehicle detection plate at the common-vehicles gate (10A).

**Figure 7 sensors-20-04131-f007:**
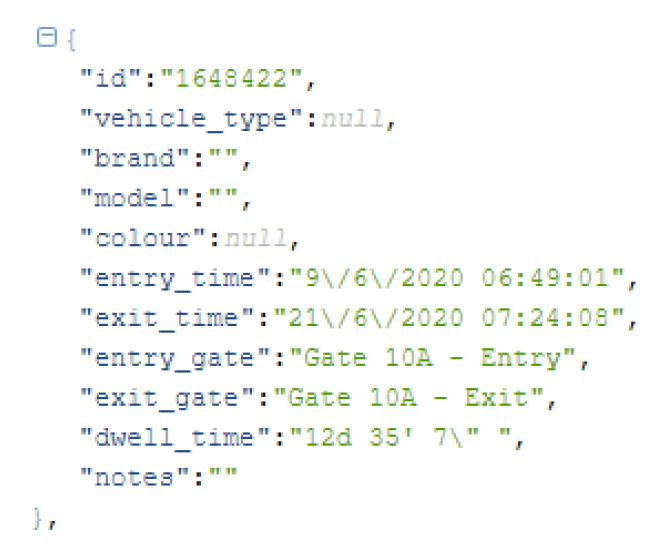
Data format of vehicles crossing the port gates.

**Figure 8 sensors-20-04131-f008:**
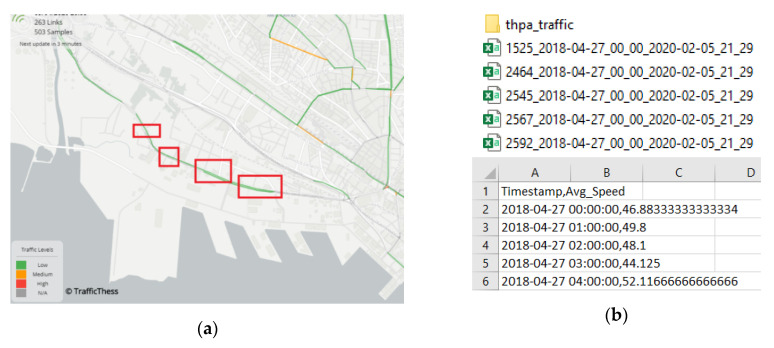
Traffic city data origin and procedure is shown as a figure, (**a**) Selected relevant surrounding links over the TrafficThess web interface; (**b**) Structure and management of historic data in CSV format.

**Figure 9 sensors-20-04131-f009:**
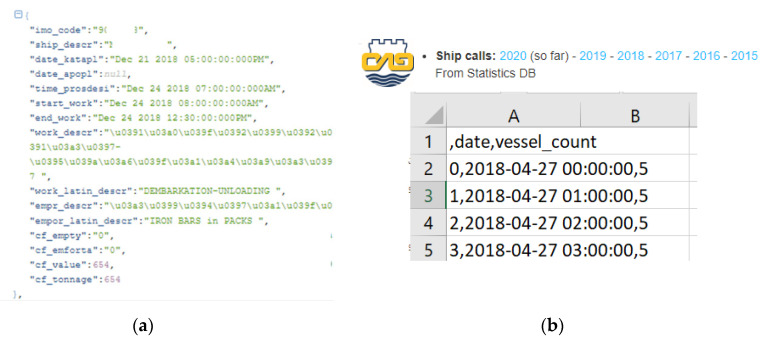
Vessel count per hour in the Port of Thessaloniki: (**a**) Raw data format; (**b**) Structure and management of historic CSV.

**Figure 10 sensors-20-04131-f010:**
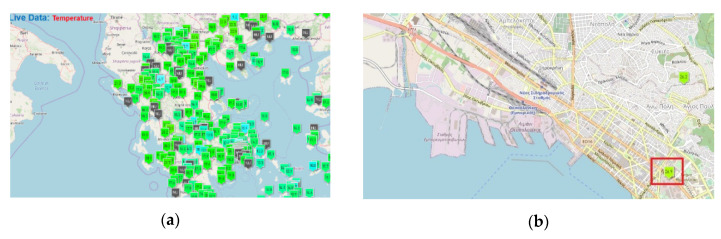
Selection of sensor data source for weather. (**a**) Global meteo sensor stations distributed throughout Greece; (**b**) Sensor selected close to the gates of the Port of Thessaloniki.

**Figure 11 sensors-20-04131-f011:**
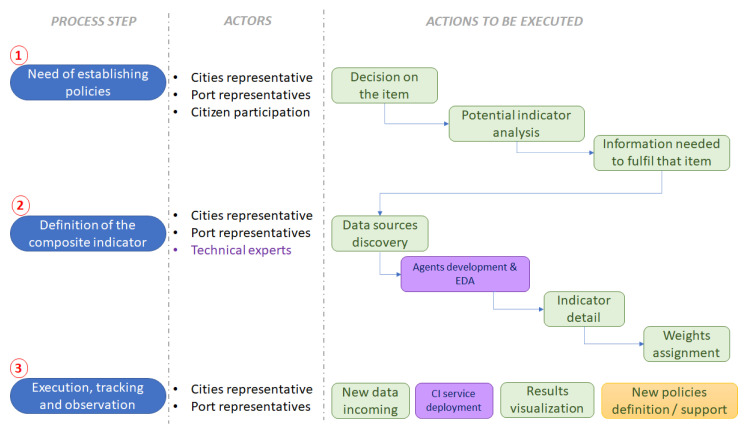
Conceptual methodology for deploying the framework on a practical experiment.

**Figure 12 sensors-20-04131-f012:**
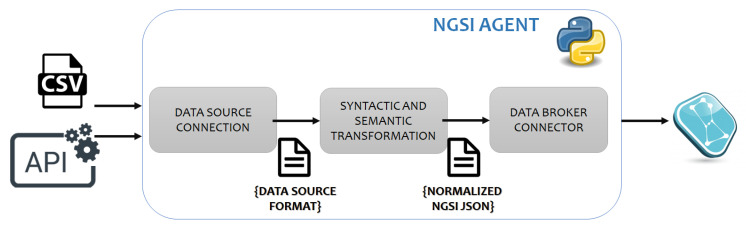
NGSI agent internal process for data conversion in the TCI experiment.

**Figure 13 sensors-20-04131-f013:**
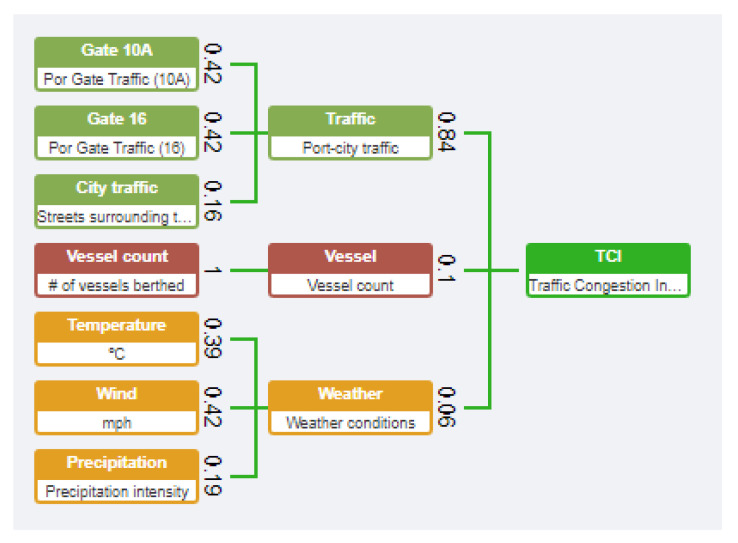
Tree definition and weighting for calculating the Traffic Congestion Index.

**Figure 14 sensors-20-04131-f014:**
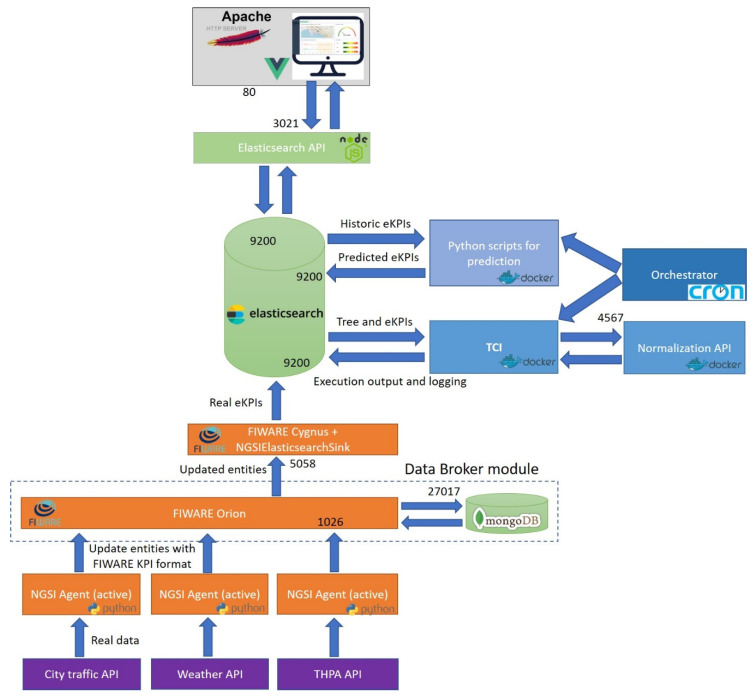
Integration and deployment of the framework for running the experiment.

**Figure 15 sensors-20-04131-f015:**
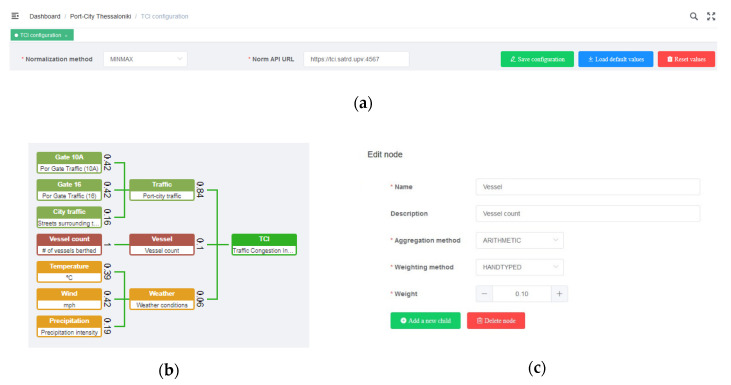
Composite indicator configurations: (**a**) normalization method and API; (**b**) tree structure creation, from results of Section 3.4.2; (**c**) configuration for each node of the TCI tree.

**Figure 16 sensors-20-04131-f016:**
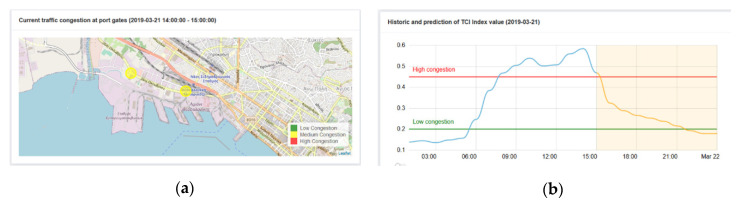
Configuration of thresholds: (**a**) for the representation of current status in a map; (**b**) for visualizing margins of TCI evolution and prediction.

**Figure 17 sensors-20-04131-f017:**
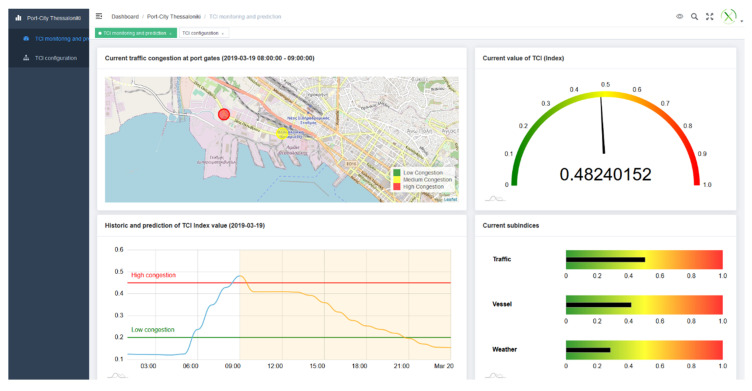
UI dashboard to visualise and interpret the calculation and prediction of the TCI.

**Figure 18 sensors-20-04131-f018:**
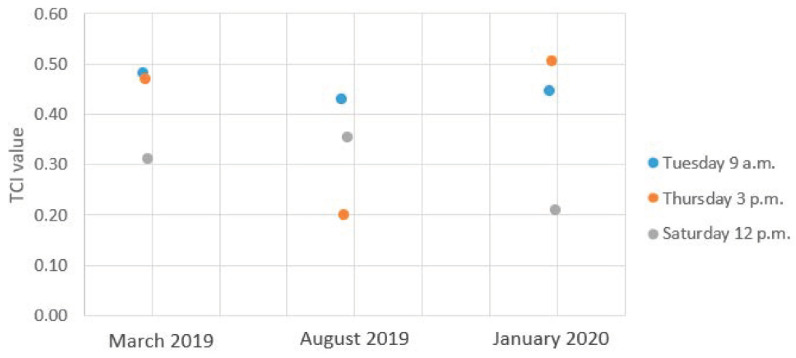
Results of the experiment—graph of TCI values per scenario.

**Table 1 sensors-20-04131-t001:** Technical and functional requirements for designed the IoT framework of this paper.

Requirement	Type ^1^	Coverage by Module
Short- and long-term storage of the data	Technical	Data storage
Automatic scheduled execution	Technical	Orchestrator
Flexibility for filtering, selecting the data	Technical	Data storage
Semantic interoperability and common syntax	Technical	Data Broker, agents
Agile integration and deployment	Technical	Containerisation
To be able to add new data sources	Functional	Data Broker, agents
To train a predictive model and use it for inferring CI	Functional	Training, inference
To setup the weights and methods to calculate the CI	Functional	CI, orchestrator, UI
To visualise the current value of the indicator (real-time)	Usability	UI – User interface
To visualise the evolution of the indicator during a day	Usability	UI
To visualise the predicted evolution of the indicator	Usability	UI, training, inference
To make the framework as configurable as possible	Usability	UI

^1^ Here, authors did not consider exploitability and non-functional requirements, neither used specific tools or methodologies to come up with the table above. Requirements were extracted from authors’ previous experiences and from concerns witnessed by contact with close port-city stakeholders.

**Table 2 sensors-20-04131-t002:** Representation of a smart port-city composite index use-case.

Event to Model	Actor Intervening	Relevant Data	Further Usage of Index
Traffic congestion at the interface of the port with the city—TCI	Port Authority	- Traffic at the gates of the port - Vessels berthed at the port	Internal process- and relationship-centric practices, leading to less traffic and pollution.
	City Municipality	- Traffic in the city - Weather in the city	Monitoring and auditing.
	Public/citizens		Knowledge and port acceptance.

**Table 3 sensors-20-04131-t003:** Summary of data used in the experiment.

Data Source	Technology Used	Relevant Parameters	Refresh Frequency of Pre-Processed Data	Units	History Available
Gates’ traffic	RFID	Vehicles	Per hour	Vehicles/hour	April 2018
Weather	Meteo station	Temp, wind speed, precipitation intensity	Per day	°C, kmh, mm	September 2018
City traffic	GPS	Average speed	Per hour	Kmh/vehicle	April 2018
Vessels in port	Vessel calls	Time ranges	Per hour	# of vessels	April 2018

**Table 4 sensors-20-04131-t004:** Interoperability NGSI agents developed in the experiment.

Data	Agent Type	Format	Transformation	RT	kpiName
**Gates traffic—**Historic	Active over static file	CSV	Pre-processing explained in Section 3.3.	-	Kpi-traffic-gate-10A/16
**Gates traffic—**real-time	Active over dynamic source	REST API	Filtering, grouping and JSON conversion.	60′	Kpi-traffic-gate-10A/16
**City traffic**	Active over external website	CSV	Pre-processing explained in Section 3.3	60′	Kpi-traffic-city
**Weather**	Active over external website	CSV	Pre-processing explained in Section 3.3	24 h	Kpi-weather-temperature -windSpeed -precipIntensity
**Vessel count—**historic	Active over static file	CSV	Pre-processing explained in Section 3.3	-	Kpi-vessel-count
**Vessel count—**real-time	Active over dynamic source	REST API	Filtering, grouping and JSON conversion.	60′	Kpi-vessel-count

**Table 5 sensors-20-04131-t005:** Correlation between all data sources with the traffic at the gates of the port.

Source	Correlation	Explanation
Average speed in nearby streets	−19.2%	As the cars move slower (less speed), the more traffic at the gates is experienced, being the most statistically relevant factor influencing the congestion
Number of vessels berthed	+11.7%	More vessels at the berth, more traffic at the gates. Although the correlation is not high, this % is significant.
Temperature	−7.7%	The colder, the more traffic, which under a logical point of view: summer-less traffic. Not much statistical impact.
Wind speed	−8.6%	The winder, the less traffic, but with no relevant relation to be conclusive.
Precipitation intensity	+3.7%	Very loosely coupled, almost no statistical correlation.

**Table 6 sensors-20-04131-t006:** Hardware specifications of the server.

Item	Specifications
CPU	4 CPUs x Intel^®^ Xeon^®^ CPU E3-1220 v5 @ 3.00 GHz
Storage Memory	HDD 100 GB
RAM Memory	16.05 GB
Cluster	FUJITSU PRIMERGY TX1330 M2

**Table 7 sensors-20-04131-t007:** Software specifications of the framework.

Item	Specifications/Version
Server OS	Ubuntu Server 18.04.4 LTS
Java	OpenJDK 1.8.0_252
Apache Maven	3.5.4
Node.js + npm	12.18.1 LTS + 6.14.5
Python + pip	3.8.3 + 20.1.1
Docker + Compose	18.09.7 + 1.17.1
Apache Server	2.4.43
Elasticsearch	7.8.0
MongoDB	3.6.3
FIWARE Orion	3.4.0
FIWARE Cygnus	2.2.0
Development IDEs	Eclipse IDE for Enterprise Java Developers (v. 2020-06) Visual Studio Code (version 1.46)

**Table 8 sensors-20-04131-t008:** Experiment results, reasoned per scenario.

Sc.#	Date	TCI Value	Explanation and Evolution
1.1	19 March 2019 9 a.m.	0.48240252	Peak is observed at the hour of the scenario execution, which is aligned with the EDA and the expected situation. The curve is a bit more flattened than logically expected, but downwards timing seems proper.
1.2	21 March 2019 3 p.m.	0.46978970	The hour of the scenario is exactly experiencing the start of congestion dawn, where the peak was reach just before 3 p.m. The prediction curve looks proper.
1.3	23 March 2019 12 p.m.	0.31322503	Scenario 1.3 can be trusted as well as the prediction for the central hours (9 to 15) experiences usual up and downs and it is kept between mid-congestion margins.
2.1	13 August 2019 9 a.m.	0.43149143	As a usual Tuesday, levels of congestion remain constant high levels during the labor days. Curve reflects with high accuracy the usual picture in vessel-operations busy months.
2.2	15 August 2019 3 p.m.	0.20111357	Scenario 2.2 shows unusual representation: despite being a Thursday, the traffic congestion is experienced and predicted between 0.2 and 0.3. However, this must be considered nothing but a good functioning of the framework and the prediction, as the 15th August is national holy day in Greece.
2.3	17 August 2019 12 p.m.	0.35446125	Scenario 2.3 follows the same rationale than 1.3, therefore it is valid. The only addition is that more traffic congestion (in general), is perceived and forecasted. This makes sense as 17th August is summer period and, normally, more traffic is experienced in the city on Saturdays at central hours.
3.1	14 January 2020 9 a.m.	0.44768116	Same exact observation, thus rationale than 2.1.
3.2	16 January 2020 3 p.m.	0.50527066	As in 1.2, hour of experiment coincides with congestion dawn. Curve looks legit.
3.3	18 January 2020 12 p.m.	0.21129410	Scenario 3.3 registered one of the lowest TCIs, both at the measurement hour and in the real-time previous values and the forecasted (max. 0.3). This makes sense as winter’s Saturdays are less congested (in general) than the rest of the days. City’s weekend tourist life is not as vibrant as at summer.

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
