# Peer review of "Framework and Methodology for Establishing Port-City Policies Based on Real-Time Composite Indicators and IoT: A Practical Use-Case"

_sensors, 2020, doi:10.3390/s20154131_

Round 1

Reviewer 1 Report

Page 2: Please use the full name together with the acronym (i.e. Association Internationale Villes et Ports – AIVP, Non-Governmental Organization - NGO,   Information and Communications Technology -  ICT,  Internet of Things - IoT). I believe that there will be some readers who are not familiar with these acronyms.  The same also applies in the case of page 13: RFID - Radio-frequency identification, page 14: GPS - Global Positioning System.

Page 3: “However, it was also observed that there are several data types from ports that will never be opened to the public like those that mainly concern business model operations or private protected data.” I believe that you refer more to the “researchers” and other stakeholders than to the general “public”.  Please clarify.

Page 6: “For example, traffic congestion or other transportation/mobility services produce harmful emissions to the atmosphere (NOx, CO2, PMs, etc.) while…”. Please note that “traffic congestion” is not a transportation/mobility service. Please correct accordingly.

Page 12: “….the aforementioned port activities and growing businesses is hugely impacting the traffic congestion in the surrounding streets to the port and city center…..”. Please provide full justification concerning the reasons why port activities “hugely impacting the traffic congestion in the surrounding streets to the port and city center”. I would like to see the total daily traffic volumes together with the traffic composition in the impact area of the port and then the % of these volumes which are associated to the port activities.

The same applies in the case of “….The main truck gate of the Port is on its west side and imposes a significant amount of traffic during the peak hours…..”. Peak hours for the general traffic or peak hours for the port? These are not always coincided.

Page 12: “….However, there are other factors influencing the congestion, such as port activities (e.g. by the number of ships being operated), the weather conditions….”. Can you please provide some more details about why “weather conditions” influence congestion in the case of the gates of the port (in addition to the text provided on page 16) ?

Page 25, Table 8. Experiment results, reasoned per scenario: Scenario 3.3.” This makes sense as winter's Saturdays are less congested (in general) than the rest of the days. City's life is not as vibrant as at summer.”. Please note that during summer period (especially during July and August) the city is rather “empty” since too many residents are on holidays. Therefore, the road network examined (western port gates) is heavily congested during winter time and not during summer time.  Can you please check scenario 2.3 for example ?

Reviewer 2 Report

The article shows an application of different IT methods and tools aiming to produce Smart Port-City's Composite Indicators with the main purpose of monitoring urban and maritime traffic congestion. The methodology and outcome seem to be interesting and useful to solve some logistical decision-making problems.

Nevertheless, I suggest to improve the overall paper's readability and fluency through some english changes, and revising/replacing a couple of concepts in the text. Finally, the authors could communicate the novelty of the approach and its utility more clearly within the first part (introduction) and the conclusions.
I list here some suggestions line-by-line.

1.INTRODUCTION

Page 2 (Lines 68-70) [While complicated to measure...] - This sentence is phrased a bit confusingly, and its meaning is not very clear. Please, consider revising it.

Page 2 (Lines 72-74) [However, today, there are no relevant indicators...] - Could you add a citation or provide some evidence about this sentence?

Page 2 (Line 83) ["own features"] - Are the quotation marks really necessary here?

Page 2 (Lines 84-86) [To establish policies...] - I'm not sure if the sentence's grammar is correct. Anyway, could you try to better stress the importance of indicators for policies? (Which type of policies? And Why?). It seems that this issue is one of the cores of your paper. Maybe, you could be more accurate here.

Page 2 (Lines 101-109) [The contributions made in...] - These are the paper's goals. You could better introduce them, also emphasising the relevant issues through some research questions to which you are trying to respond with your methodology. Just as suggestion about the list of goals, you could use this form: "The paper's objectives proceed as follows: 1) ....;2)...,etc..."; and you should consider to revise all the sentences in the list.

Page 3 (Lines 112-113) [The authors believe...] - I suggest to move this sentence to the conclusions section or delete it. It's a bit vague.

Page 3 (Line 115) [Application, deployment,...] - This sentence needs the verb.

Page 3 (Line 123) Just as suggestion, this sentence could be: "The reminder of the article proceeds as follows".

2.RELATED WORK

Page 3 (Line 141) [dyadic application] - I'm not an expert in field, but I suggest you to specify this concept with an example in order to facilitate the overall comprehension.

Page 4 (Line 199) ["official"] - The term is redundant, please check the misprint.

Page 5 (Line 242) ["[47] explains that..."] - Please, consider revising this sentece since it is not clear.

Page 7 (Line 329) - You could replace "resilincy" with "resilience" here.

3.MATERIALS AND METHODS

Page 13 (Lines 558-578) [For a port authority...] - Maybe you can move this sentences to the conclusions and discussion section, after expanding them.
You should try to better stress how the community has been involved in this project.
This seems to be the core of the paper. I suggest to better stress the potentials of proposed methodology and explain broadly the concepts of democratisation of information within the decision-making problems which involve citizens and port-authority.

Page 13 (Lines 581-583) [The paper does not aim...] - It's vague. I suggest to move it to the conclusions, after expanding and stressing it, or you could consider deleting this sentece.

Page 19 (Lines 776-787) [Weighting: The last-but-one...] - You could spend more words about the description of the weighting procedures since it is a crucial step for deriving outputs. Moreover, the aggregation procedure has not been well explained. Which type of arithmetic aggregation has been chosen? Please, clarify it deeply.

5.Conclusion and Future Research Lines

You should extend the conclusions outlining some limitations of the proposed approach. The conclusions are a bit short and vague. Moreover, you could expand them with deeper consideration about the novelty of the research, highlighting other potentials referring to the software application. Which type of decision-making problems the method could solve? And why?

Author Response

Please see the attachment. Cover letter answering the questions is placed after the end of the manuscript.

Round 2

Reviewer 2 Report

The authors solved all the comments and added other sentences to try to better underline some confusing concepts. Nevertheless, about new parts and the overall paper, I remark that an extensive English proof-reading can aid to further clarify the design method and to enhance the quality of presentation.

Additional remarks and suggestions follow:

Line 54. Please, replace "pollutive" with "polluting"

Line 64. Please, replace "longer-term" with "long-term"

Line 108. Please delete "on that connect" since it is ambiguous.

Line 119. Please, delete "tentative" and "of this paper" since the sentence is a bit phrased.

Line 131. Replace "Timeseries" with "Time-series"

Line 153. You could delete "among others" at the end of brackets

Line 210. You could replace "tentative" (obj.) with "attempt" (noun)

Line 251. Please replace "sub-set" with "subset" (without a dash)

Line 500. What the term "dockerised" means? I cannot find this word.

Line 529. Maybe "contextualized" can be replaced with "determined by" or similar terms.

Line 553. You could delete the round brackets here

Line 557. Maybe "latite" stay for "attitude"? Please check the misprint.

755-763. The explanation is not much understandable due to its grammar.

769-780. These sentences are very confusing. I can understand with difficulty the meaning. A grammatical revision needs here.
